# Multiplexed Nanobiosensors: Current Trends in Early Diagnostics

**DOI:** 10.3390/s20236890

**Published:** 2020-12-02

**Authors:** Greta Jarockyte, Vitalijus Karabanovas, Ricardas Rotomskis, Ali Mobasheri

**Affiliations:** 1Department of Regenerative Medicine, State Research Institute Centre for Innovative Medicine, Santariskiu 5, LT-08406 Vilnius, Lithuania; greta.jarockyte@imcentras.lt (G.J.); ali.mobasheri@imcentras.lt (A.M.); 2Biomedical Physics Laboratory, National Cancer Institute, Baublio 3b, LT-08406 Vilnius, Lithuania; ricardas.rotomskis@nvi.lt; 3Research Unit of Medical Imaging, Physics and Technology, Faculty of Medicine, University of Oulu, FI-90014 Oulu, Finland; 4Departments of Orthopedics, Rheumatology and Clinical Immunology, University Medical Center Utrecht, 3508 GA Utrecht, The Netherlands

**Keywords:** multiplexing, nanoparticles, quantum dots, gold nanoparticles, silver nanoparticles, upconverting nanoparticles

## Abstract

The ever-growing demand for fast, cheap, and reliable diagnostic tools for personalised medicine is encouraging scientists to improve existing technology platforms and to create new methods for the detection and quantification of biomarkers of clinical significance. Simultaneous detection of multiple analytes allows more accurate assessment of changes in biomarker expression and offers the possibility of disease diagnosis at the earliest stages. The concept of multiplexing, where multiple analytes can be detected in a single sample, can be tackled using several types of nanomaterial-based biosensors. Quantum dots are widely used photoluminescent nanoparticles and represent one of the most frequent choices for different multiplex systems. However, nanoparticles that incorporate gold, silver, and rare earth metals with their unique optical properties are an emerging perspective in the multiplexing field. In this review, we summarise progress in various nanoparticle applications for multiplexed biomarkers.

## 1. Introduction

The clinical diagnosis of many diseases depends on the accurate and unambiguous detection of various biomarkers, which may include proteins or other biomolecules [1,2,3]. Immunoassays are a well-established tool for measuring analytes that are normally present at very low concentrations and cannot be determined accurately by other less expensive tests. Standard immunoassays are used to detect one specific analyte, however, for complex diseases such as cancer, diabetes, rheumatoid arthritis (RA) and osteoarthritis (OA), which involve a multitude of biomarkers, one analyte is not enough to obtain an early and accurate diagnosis. In this case, multiple immunoassays can be employed. However, immunoassays require samples that are collected, labelled, archived, and bio-banked according to established laboratory protocols before even conducting the assays, which makes this method challenging and time-consuming. In order to diagnose at the earliest stages of disease development, it is crucial to detect as much information as possible from small quantities of clinical samples. In this context, multiplexed immunoassays are an obvious and attractive approach, especially when sample volumes are limited. The main advantage of multiplexed analysis is the capability to detect multiple biomarkers qualitatively or quantitatively in a single sample. There are additional advantages to this approach, such as more data points from single samples, reduced cost per data point, fewer errors due to fewer samples, and increased throughput (Figure 1). Conversely, cross-reactivity remains a significant challenge. If antibodies cannot distinguish between the analyte and other structurally similar components, the specificity of immunoreactivity drastically decreases, and as a consequence, multiplexed detection may not work properly in complex biological and clinical samples.

Through advances in nanomedicine, as well as progress in areas such as nanoelectronics, nanomaterials, and optics, there has been a steady evolution in the development of miniaturised nanobiosensors for performing biological analysis based on a variety of multiplexing technologies. Nanobiosensors are sensors in which various nanosize materials are used, however, large equipment is still required for whole sensing systems, because they are not, by themselves, able to detect nanoscale events. The current state-of-the-art nanobiosensors are based on several diagnostic methods, such as enzyme-linked immunosorbent assay (ELISA), flow cytometric immunoassay, electrochemical (amperometric, voltammetric and impedance), immunoassay, mass spectrometric immunoassay, chromatographic immunoassay, and different optical immunoassays. Contemporary multiplexed nanobiosensors are developed for the detection and quantification of clinically relevant analytes in biological and clinical samples (i.e., blood, urine, and saliva).

However, in order to successfully implement multiplexed biosensing systems, several crucial elements are required [4,5]. The two key steps in any immunoassay are efficient capture and specific detection. Firstly, recognition probes (e.g., antibodies and other biomolecules), that can interact with specific analyte are required. Specific antibodies can recognize analytes, however for the detection of immunoreactivity, additional elements are needed. Thus, a second and crucial factor is required—the detection probe. For this purpose, stable optical markers or labels that can provide the signal contrast in the final imaging process can be used. These probes with bio-functional groups provide optical signals to confirm and quantify the specific analytes. The last element is a readout system, which can evaluate and quantify detection probes, thus providing information about specific analytes. Signals for readout are usually based on fluorescence parameters such as wavelength, intensity, and lifetime. Although a variety of different fluorescent dyes and fluorophores have been used as detection probes in various immunoassays, only a few fluorescent probes are capable of monitoring multiple analytes in bioassays with distinct optical signals. Compared to fluorescent dyes, nanoparticles are stable, bright fluorophores with high fluorescence quantum yields, narrow fluorescence emission bands (quantum dots and upconverting nanoparticles), larger Stokes shift, long fluorescence lifetimes, high resistance to photobleaching, and can provide excitation of several different emission colours (quantum dots) using a single wavelength for excitation. In this review, we summarize progress in nanomaterial-based multiplexed bioassays, by focusing on their applications in medicine.

## 2. Quantum Dots

Since their first description in a biological context [6] quantum dots (QDs) have been considered one of the most attractive luminescence label in biomedicine. Quantum dots are inorganic semiconductor crystals with a diameter of 1 nm to 10 nm (typically 2–6 nm), with optical properties which arise from interactions between electrons, holes, and their local environments [7,8]. QDs absorb photons whose energy exceeds the photoluminescence excitation bandgap. In this process, electrons are transferred from the valence band to the conduction band. Due to many electronic states, QD photoluminescence excitation spectra are wide, but on the other hand, QD emission bands are narrow and symmetrical. These unique properties allow different QDs to be excited by the same light source and obtain different coloured visualisations in one biological sample. For comparison, the majority of organic fluorescent dyes have relatively narrow excitation bands and wide fluorescent band. Therefore, in order to perform multicolour imaging, several different light sources are required [8]. By varying the size and composition of QDs, the emission wavelength can be tuned from blue to near-infrared [8]. Additionally, because the surface of QDs are easily modifiable, it is easy to attach various antibodies and other biomolecules to them [9]. Thus, QDs are highly applicable for multiplexed immunoassays. Here, we briefly review how QDs could be used for various multicolour biosensing applications.

### 2.1. QLISA

Quantum dot-linked immunosorbent assay (QLISA) is an approach similar to the enzyme-linked immunosorbent assay (ELISA), except QDs are used instead of enzymes [10]. The principle of these two methods is the same. For both methods, the first step is antibody coating—the specific capture antibody is immobilised on high protein-binding plates. When the capture antibody is immobilised, the uncovered space on the plate is blocked with an irrelevant blocking protein, e.g., serum albumin. After that, samples and standard dilutions are added to the wells. In this step, the analyte adheres to the capture antibodies. In some ELISA cases, analytes can be attached directly on the plate surface, without using a capture antibody. The excess sample is removed, and then specific biotinylated detection antibody is added to the wells to enable detection of the captured protein. In the case of QLISA, QDs with conjugated detection antibody are used. After detection, the antibody binds to the analyte, and streptavidin conjugated with alkaline phosphatase/horseradish peroxidase (ELISA) or streptavidinated QDs (QLISA) is added to the wells which binds to the biotinylated antibody. For the ELISA method one addition step is required before analysis—addition of colorimetric substrate which form a coloured solution when catalysed by the enzyme. Samples are then analysed with microplate readers. The optical density in each well of the multi-well plate is measured, whereas for QLISA, photoluminescence of the QDs is registered.

ELISA is an established standard biomarker detection method that is widely used in biomedical research and in the clinical diagnostic setting [2,11], whereas QLISA is still in the development stage. Nevertheless, the ELISA method still has some important disadvantages, which hopefully QLISA could solve. The main drawback of the conventional ELISA method is that the detection limit is barely less than the nanomolar concentration level, which is usually not enough to reach the clinical threshold of many protein biomarkers, especially in the early stages of disease [12]. This disadvantage is due to low signal-to-noise ratio and limited signal amplification [13]. An additional drawback is that traditional ELISA detects only one analyte in a sample, and multiplexed detections require several measurements simultaneously carried out [14]. These disadvantages can be solved by using luminescent QDs instead of enzymatic colorimetric reactions for identification of the detection antibody. First of all, photoluminescence-based assays are more sensitive than absorbance-based assays [15], thus the signal-to-noise ratio increases by measuring photoluminescence instead of absorption. Moreover, QDs have a high photoluminescence quantum yield, which allows acquisition of high signal intensities from relatively low concentrations [16]. Therefore, QDs could work as amplifiers when low concentrations of analyte need to be detected. Finally, QDs, due to their optical and chemical properties, are good candidates for use in multiplexing (Figure 2).

Mansur et al. (2015) successfully demonstrated the detection of the cluster of differentiation 20 (CD20) antigen, which is a biomarker overexpressed by B-lymphocyte cancer cells, using QLISA [17]. Suzuki et al. (2017) reported the application of QLISA for the detection of the multifunctional cytokine interleukin 6 (IL-6) which is involved in many cellular processes, such as receptor synthesis, inflammation, cell proliferation, and cancer cell signalling [10]. Functionalised QDs were used in QLISA to detect and quantify IL-6. Results demonstrated that the lower limit of IL-6 detection would be approximately 50 pg/mL, which is undetectable using a standard ELISA method [10]. However, these two studies demonstrated working QLISA for the detection of only one biomarker. Nevertheless, the strategies that they described could also be applied for multiplexed QLISA by increasing the number of different QDs-antibody conjugates to detect more than one biomarker.

The first multiplexed QLISA was described by Goldman et al., published in 2004 [18]. In this study, four different toxins (cholera toxin, ricin, shiga-like toxin 1, and staphylococcal enterotoxin B) were simultaneously detected using CdSe/ZnS QDs (510, 555, 590, and 610 nm, respectively) bioconjugates. The detection of the four analytes was demonstrated at 1000 and 30 ng/mL of each toxin in the mixture. In their study, Goldman et al. demonstrated the principle of multiplexed QLISA and proposed that their work highlights the advantages of QDs versus conventional organic dyes for use in multiplexing applications. However, additional studies are clearly necessary to optimise assay conditions and antibody reagents to produce a reliable and robust multiplexed assay [18].

Song et al. (2015) demonstrated the application of QLISA for the detection of multiplexed residues of antibiotics in milk [19]. A CdTe QDs detection probe conjugated with different antibodies against antibiotics was used. The limit of detection of this assay is 5 pg/mL, thus the developed method showed excellent sensitivity and specificity for the target analytes. Moreover, this assay is quicker than standard ELISA and can be performed in 90 min. Although the authors focused on developing multiplexed nanobiosensors for food safety control, Song et al. developed a method that can easily be adapted for the purpose of various biomarkers detection.

### 2.2. Magnetic Bead-Quantum Dot Assay

Magnetic beads are usually used for the extraction or purification of biomolecules such as proteins, antibodies, and nucleic acids. Bio-functionalised magnetic beads are widely used in cell sorting, bioseparation, and immunoassays. Functionalisation of magnetic beads with specific antibodies leads to decent recognition, separation, and collection of biomarkers in liquid biopsies without additional separation procedures. High surface-to-volume ratios of magnetic beads enables the capture of analytes in a low concentration solutions [20].

The first report of a multiplex magnetic bead assay was in 1977 [21]. This was an immunoassay that used magnetic beads to simultaneously measure multiple analytes in a clinical sample. A multiplex magnetic bead assay is a derivative of a traditional ELISA, using beads for binding the capture antibody, which are available in several different formats based on the utilisation of flow cytometry, chemiluminescence, or electrochemiluminescence technology [14,22]. As mentioned earlier, a traditional ELISA detects only one analyte in a sample, whereas this method was developed with the purpose of measuring multiple analytes in the same sample at the same time. Confusion can be caused by the fact that this method has been described in various ways in the literature, as multiplex bead array assays [14], Luminex multi-analyte or Luminex assay (trademark of Luminex company), multiplex ELISA [23], magnetic immunoassay [24], flow cytometric multiplex arrays or bead-based multiplex assays [22], or a cytometric bead array system (product of BD Biosciences). All the methods mentioned in this context use different antibodies for efficient capture of analytes, while magnetic beads separate target biomolecules from other molecules in the clinical specimens. Nonetheless, bioassays differ according to the optical detection systems used and the different terminologies used in the scientific publications.

There are several multiplex magnetic bead assays, which are different from each other only by virtue of small modifications. For clarity, we will shortly explain the most commonly used method—the Luminex assay. This immunoassay technique is the ELISA with flow cytometry [25]. For this assay, paramagnetic microspheres or beads that are internally labelled with different concentrations of fluorophores (red and infrared) and conjugated to analyte-specific capture antibodies are used. Each fluorophore combination corresponds to one analyte. A mixture of colour-coded beads is added to the sample (or vice versa) and the antibodies bind to the analytes of interest. By completion of this procedure, the additional complementation of biotinylated detection antibodies leads to the formation of an antibody–antigen sandwich. At this point, fluorophore (commonly phycoerythrin (PE)) labelled streptavidin is added, which binds to the biotinylated detection antibodies. For detection, a dual-laser flow-based detection instrument is used for bead classification, analyte determination, and amount of analyte evaluation. This assay can identify multiple biomarkers in biofluid samples and precisely quantify them [25].

Magnetic bead–quantum dot assay is a similar detection method, except instead of organic dyes, quantum dots are used (Figure 3). There are several articles that have established magnetic bead–quantum dot assays to detect one specific analyte [20,26,27,28]. The most sensitive method using a magnetic bead–quantum dot sandwich assay for the capture and detection of human S100B protein (most extensively studied biomarker for mild traumatic brain injury) in serum was demonstrated by Kim et al., in 2015 [20]. Kim et al. have demonstrated that their method provides highly specific detection of S100B protein—with a detection limit of 10 pg mL^−1^—and a 3 order of magnitude of the detection range up to 10 ng mL^−1^. The overall assay time is 1 h, making this method relatively fast and sensitive [20].

In 2017, the same group developed a droplet-based device for simple, fast, and cheap malaria biomarker PfHRP2 detection [29]. This method was designed by adapting a magnetic bead–quantum dot assay to vial-based assay. This detection tool allows for quantitative and sensitive evaluation of the target biomarker [29].

Liu et al. (2016) proposed multiplex magnetic bead–quantum dot assay in microarray format to detect lung cancer biomarkers [30]. The 6.5 μm diameter magnetic beads and QDs (525 nm, 585 nm and 625 nm) both conjugated with antibodies against cytokeratin-19 fragment (CYRFA 21-1), neuron specific enolase (NSE) or carcinoembryonic antigen (CEA) lung cancer biomarkers were added into human serum samples and mixed in a test tube, and were introduced onto the array of the chip. Microbead–antigen–QD conjugates were detected by fluorescent microscope, equipped with a charge-coupled device (CCD) camera. Results showed that this sandwiched immunoassay successfully detects the presence of different lung cancer associated biomarkers such as CYRFA 21-1, NSE and CEA in the given biological fluid at low concentrations (detection limit 0.97 ng/mL; 0.37 ng/mL; 0.19 ng/mL, respectively). The authors suggest that this method could be a low cost tool for disease diagnosis [30]. In later work, the same group demonstrated a similar approach focusing on the same three lung cancer biomarkers, using the multiplexed detection and micro-magnetic beads as immune carriers and QDs as detection probes [31]. Immunocomplexes were visualised using fluorescence microscopy and the micrographs were analysed by image analysis software to quantify the luminescence of QDs. Using this method, the authors managed to reach lower than 1 ng/mL detection limit (364 pg/mL for CYRFA 21-1, 38 pg/mL for CEA, 370 pg/mL for NSE) [31].

Bai et al. (2019) used bead-based microarray to detect three lung cancer biomarkers (CEA, CYFRA 21-1 and ProGRP) from exosomes [32]. By using a microfluidic system, exosomes were collected from cell culture supernatant or plasma samples from lung cancer patients. After the isolation of exosomes, tumour biomarkers were detected by using QDs conjugated with detection antibodies. The difference between experimental result and clinical data was minimal, thus this method could be applied for clinical testing [32].

Li and co-workers developed barcodes for multiplexed detection using magnetic beads and QDs [33,34]. In 2016 they introduced barcodes for the multiplexed detection of five different tumour biomarkers in serum (AFP, CEA, CA199, CA125, and CA242). Cadmium-free NIR-emitting CuInS_2_/ZnS QDs and superparamagnetic Fe_3_O_4_ in PSMA microspheres were used for simultaneous biomarkers detection. Using these barcodes, they detected sub-ng/mL concentrations of analytes [33].

### 2.3. Multiplex Flow Cytometric Immunoassay

Magnetic beads are convenient for immunoassays, due to the possibility of controlling them with a magnetic field and separating them from the mixture when it is needed. However, non-magnetic beads are more frequently used for multiplexed flow cytometric immunoassays.

Yu et al. (2012) have developed a competitive microbead-based flow cytometric immunoassay for the simultaneous detection of two biomarkers using QDs luminescent labels [35]. The purpose of their study was the development of a duplex detection system for the detection of toxins (microcystin-LR, benzo[a]pyrene), which can be found in drinking water as contaminants. These two molecules are too small to apply a sandwich immunoassay, thus a competitive immunoassay was designed to determine molecules using a single antibody. During competitive reaction, antigen competitors on polystyrene beads are changed with target molecules. Then, QDs conjugated with antibodies against the target molecules are added to the solution. The final stage of the assay is photoluminescent measurement of the sample without any wash step, using a flow cytometer. QD–antibody conjugates label both free and polystyrene bead attached antigens, although during flow cytometer analysis the groups are easily separated. This method allows not only qualitative, but also quantitative analysis. Microcystin-LR detection dynamic range was 0.52–30 µg L^−1^ and for benzo[a]pyrene it was 0.13–10 µg L^−1^. The proposed assay was performed within 30 min, thus this is a relatively fast method for multiple analyte detection [35].

Bilan et al. (2017) demonstrated the detection of multiple lung cancer biomarkers using microspheres encoded with QDs (QDEMs) in clinical samples of bronchoalveolar lavage fluid [36]. In their study, 4.08, 6.1 and 8.24 μm carboxylated melamine formaldehyde resin microspheres as matrix cores and CdSe/ZnS QDs (λ_em_ = 515 nm) as optical codes for the preparation of QDEMs were used. Antibodies against three lung cancer biomarkers (AMBP, PRDX2, and PARK7) were conjugated with different sized QDEMs. Using QDEM–antibody conjugates, all three lung cancer biomarkers were successfully and simultaneously detected in clinical samples using a flow cytometer. The authors compared their method’s reproducibility and reliability with the commonly used Luminex xMAP^®^ bead-based immunoassay. Their results showed that QDEM technology may be considered as an alternative to Luminex xMAP^®^ for the diagnostic purpose using conventional flow cytometers. However, the authors did not achieve a high analytical sensitivity at low concentrations of the biomarkers, thus the sensitivity of this method remains unknown [36].

The main issue with the flow cytometry method is difficulties in achieving nanoparticles of homogeneous size. Heterogeneity of nanoparticles leads to interference due to morphology dependant on Rayleigh scattering signals.

### 2.4. Electrochemical Immunoassay

Electrochemical immunoassay is based on a solid phase system where an antibody–antigen reaction occurs and an electrochemical detection system is integrated in the same device [37]. In this method, electrochemical sensors are physically attached to the detection probe surface. When the detection probe interacts with the target analyte, a measurable electrochemical signal appears. These biosensors have impressive detection characteristics and are recognised as one the best sensing systems [37].

The first utilisation of semiconductor nanoparticle labels for this method was demonstrated for the electrochemical DNA hybridisation assay [38,39]. Since then, only a few papers have described how quantum dots could be used in multiplexed electrochemical immunoassay. A multiplex electrochemiluminescence immunoassay has been developed for the simultaneous determination of two different tumour biomarkers (alpha-fetoprotein (AFP) and CEA) in human serum and saliva, using multicolour QD labels and graphene as a conducting bridge [40]. Streptavidin-coated CdSe/ZnS (525 nm and 625 nm) were conjugated with biotin-labelled secondary antibodies (anti-AFP_2_ and anti-CEA_2_, respectively). QDs conjugated with secondary antibodies produced electrochemiluminescence reactions after the immunoreaction and the intensity of electrochemiluminescence was amplified using graphene as a conducting bridge. The quantity of AFP and CEA was indicated by the electrochemiluminescence responses of QDs_525_ and QDs_625_, respectively. The working range of the proposed method was 0.001–0.1 pg/mL and the detection limit was 0.4 fg/mL for both analytes [40].

Ultrasensitive electrochemiluminescence immunoassays have been developed for the simultaneous determination of two biomarkers, the cancer antigens CA 125 and CA 15-3. The immunosensor designed in this study can be used for the sequential detection of CA 125 and CA 15-3 biomarkers with the wide detection ranges of 1 μU/mL–1 U/mL and 0.1 mU/mL–100 U/mL with very low detection limits of 0.1 μU/mL and 10 μU/mL, respectively [41].

### 2.5. Multiplex Antigen Imaging in Cells and Tissues

Conventional immunohistochemistry (IHC) is widely used as a diagnostic technique in the field of pathology, especially in cancer diagnostics. However, the standard IHC method allows the labelling of a single marker per tissue section, which is insufficient in some cases, thus multiple sections of same tissue are stained and examined, which is time and resource consuming. The use of several QDs conjugated with specific antibodies enables the easy labelling of multiple biomarkers in one specimen and the distinguishing of cancer cells from healthy cells (Figure 4).

The first report of QD-based imaging of biomarkers was published by Wu et al., 2002 [42]. The authors used CdSe/ZnS QDs (λ_ex_ 535 nm and 630 nm) to simultaneously label two breast cancer biomarkers in SK-BR-3 cells (cell surface antigen Her2 and nuclear antigens). They demonstrated that QDs were effective for the two-colour fluorescence labelling of distinct cellular components [42].

Yezhelyev et al. published a breakthrough multiplex cell imaging article in 2007 [43]. They demonstrated the quantitative and simultaneous profiling of five biomarkers in breast cancer cells and clinical tissue specimens, using QDs for labelling. Five different QDs (QD525, QD565, QD605, QD655, QD705) conjugated with antibodies against HER2, ER, PR, EGFR, and mTOR biomarkers, respectively, were used for multiplexed detection of these antigens in breast cancer cells (MCF-7 and BT474 cell lines) and in tumour biopsy specimens. All samples were visualised using confocal microscopy and, additionally, spectroscopic measurements were performed for quantification purposes. The authors concluded that multiplexed detection and quantification of ER, PR, and Her2 biomarkers correlated firmly with the results gained from other traditional methods such as IHC, Western blotting, and fluorescence in situ hybridisation, implying that the QD-based sensing technology is applicable for the molecular profiling of tumour biomarkers in vitro [43].

Approaching clinical application, Liu et al. (2010) reported the use of QDs for the detection and characterisation of a class of low-abundant tumour cells in Hodgkin’s lymphoma [44]. They developed multiplexed detection of four protein biomarkers (CD15, CD30, CD45, and Pax5) on human tissue biopsies. Immunostaining of tissues was performed in two steps repeating them twice: firstly, mouse and rabbit primary antibodies against Pax5 and CD30 antigens, respectively, were used to bind lymphoma biomarkers. After a washing procedure, a mixture of goat anti-rabbit QD655 and goat anti-mouse QD605 secondary antibody–QD conjugates was applied to indicate the primary antibodies. The same procedure was repeated using primary antibodies for the additional antigens CD45 (rabbit) and CD15 (mouse), followed by using secondary antibody–QD conjugates (goat anti-rabbit QD565 and goat anti-mouse QD525). Using the described immunostaining concept, antibodies from only two animal species are sufficient, so this method is cheaper than standard ones. The results indicated that a distinct QD staining pattern can be used not only to detect Hodgkin’s lymphoma, but also differentiate it from benign lymphoid hyperplasia [44].

Xu et al. published a study in 2013 focusing on QD-based IHC imaging for multiplexing cancer biomarker detection on formalin-fixed and paraffin-embedded samples of tissues [45]. In this study, QD-primary Ab conjugates were used for the multiplexed detection of survivin and EF1α in fixed tissue samples. These two biomarkers have different expression profiles in cells, thus different brightness QDs were used for each analyte. EF1α, a highly expressed housekeeping protein, was conjugated with low intensity QD530, whereas survivin was conjugated with high intensity QD620. Survivin was detected only in the cancerous tissues, but EF1α was present in both cancerous and surrounding healthy tissues. Quantification results showed that survivin expression level in the cancerous tissue was much higher than EF1α, although in the non-cancerous tissues expression levels of these two proteins were similar. Thus, these methods enable cancer diagnosis with high fidelity using two biomarkers [45].

Despite current advances, there are some limitations that need further discussion and evaluation. The main issue is the nonspecific binding of QDs, which reduces the specificity of the methodologies that are being developed. Additionally, there is still a lack of commercially available media for QD fixation, which leads to low stability on prolonged storage and the inability to retain fluorescence in stained specimens.

### 2.6. Paper-Based Biosensors

Paper-based biosensors have emerged as simple, low-cost, quick, and user-friendly diagnostical tools. There are three main types of paper-based biosensors: dipstick tests, lateral flow assays, and microfluidic paper-based analytical devices (μPAD) [46]. All three methods have their advantages and disadvantages, for example, the dipstick test is easy and fast, however often inaccurate, and analysis takes more time than the test itself. Lateral flow assays are more sensitive, commonly detecting nanograms per mL, also featuring rapid detection, although excess antibodies can lead to agglutination and inaccurate results. Generally, μPAD have inefficient sample consumption and high limits of detection, thus usually μPAD is insufficient for the analysis of low abundance biomarkers present in small concentrations in biological samples. Multiplexed detection using paper-based biosensors is still a challenge itself, although for the past few years the number of publications in this field has been growing. The variety of paper-based biosensors are wide; thus, different nanoparticles could be used as labelling agents. QDs are quite a common choice, but metal nanoparticles, carbon dots, inorganic complexes, etc., could be used as well [47].

A paper-based immunochromatographic dipstick method based on sandwich immunoassay for the two-plex detection of lung cancer biomarkers such as CEA and NSE was developed by Xiao et al. [48]. High-quality carboxylated quantum dot beads were used as detection probes and a custom-made strip reader was developed to read photoluminescence signals for quantification. Under optimal conditions, this immunoassay showed a high detection limit of 37.8 pg/mL for CEA and 42.6 pg/mL for NSE and was approved in clinical human serums with high sensitivity and specificity. Moreover, the authors’ results demonstrated that quantum bead-based immunochromatographic dipstick biosensors could be used as a routine assay for rapid diagnostics of lung cancer [48].

Hydrophobic QD-doped polystyrene nanoparticle-based lateral flow test strips were developed for the multiplexed detection of lung cancer biomarkers CYFRA 21-1 and CEA [49]. Detection of antibodies was based on a standard sandwich immunoassay. Serum samples, collected from patients, were mixed with functionalised QD-doped polystyrene nanoparticles and added onto the test strips. Immunocomplexes were formed and immobilised by the capture antibodies on the test lines. Photoluminescence of QDs was detected by a portable fluorescence strip reader for further analysis. The authors demonstrated a detection range of 1.3–480 ng/mL for CYFRA 21-1 and 2.8–680 ng/mL for CEA. The detection limits for CYFRA 21-1 and CEA were 0.16 and 0.35 ng/mL, respectively. The overall assay test time was 15 min, making it a rapid, sensitive and user-friendly method [49].

Chen et al. demonstrated the multiplexed detection of CEA and prostate specific antigen (PSA), using a paper-based immunodevice [50]. Sandwich immunoassay was used for the capture and detection of cancer biomarkers. Capture antibodies of CEA and PSA were immobilised on the same zone of the paper-based device. Then, samples of human serum were applied to the device and the biomarkers captured by antibodies. Later, detection antibodies conjugated with CdTe QDs (525 and 605 nm) were used. QDs were excited and detected simultaneously under 272 nm excitation wavelength. The linear response of both biomarkers was in the 1.0–40 ng/mL concentration range and the detection limits were 0.3 ng/mL for CEA and 0.4 ng/mL for PSA [50].

## 3. Gold and Silver Nanoparticles

One of most widely used groups of nanoparticles are synthesised from noble metals, such as gold and silver. Gold (Au) and silver (Ag) nanoparticles (AuNPs and AgNPs) are the most stable metal nanoparticles, and they have numerous attractive features such as size-related electronic, magnetic and optical properties (quantum size effect), large optical field enhancements resulting in the strong scattering and absorption of light, and their perspective applications in biomedical field [51]. Noble metal nanoparticles, which are composed of tens or hundreds of atoms and typically < 2 nm size, are known as nanoclusters. These nanoclusters have molecule-like properties due to their extraordinary physical and chemical characteristics, and they have especially attracted the attention of scientists [52]. AuNPs are known as non-toxic and biocompatible nanoparticles; thus they are frequently used in studies related to diagnostic tool development, drug delivery, novel therapeutic agents, and other medical applications [53,54]. Whereas, AgNPs have unique antimicrobial features, which allows silver nanoparticles to be used in the treatment of microorganisms such as bacteria, fungi, and viruses [55]. In this article, the application of AuNPs and AgNPs for multiplexed biomarkers detection will be briefly reviewed.

### 3.1. Plasmonic Multiplex Sensing

Gold and silver nanoparticles are exceptional due to their plasmonic properties. Such nanoparticles demonstrate unique absorption and scattering characteristics in the VIS–NIR spectral region due to photon-induced collective oscillation of their surface electrons. This coherent electron oscillation in noble metal nanoparticles is also known as localised surface plasmon resonance (LSPR). Briefly, LSPR occurs when a nanoparticle (bigger than 2 nm) is interacting with light. Electromagnetic oscillating fields cause the conduction electrons of a nanoparticle surface to oscillate coherently. A highly localised oscillating electron cloud is created around the nanoparticle, which rapidly decays away, through resonance enhance far-field scattering. The LSPR profile can be described with spectroscopically measured parameters such as shape, position and intensity, which strongly depend on the properties of nanoparticles, such as size, shape, monodispersity, as well as interaction with ligands or surrounding media [56].

Huang et al. (2012) developed a multiplexed bioanalytical assay which allows simultaneous detection of human serum specimens infected by *Schistosoma japonicum* and tuberculosis pathogens without sample pre-treatment. Two different populations of gold nanorods (AuNRs) were modified with antibodies of different antigens. Detection of biomarkers was achieved by registering a shift in the LSPR, using a standard visible/NIR spectrometer. This nanobiosensor was able to identify infected and uninfected samples and provide a semi-quantitative readout [57].

Utilising the LSPR properties of plasmonic metal nanoparticles, a new variation of the conventional ELISA was established, known as plasmonic ELISA (pELISA) [10]. Several studies have demonstrated the application of pELISA for diagnostics of diseases such as prostate cancer [58,59,60], syphilis [61] tuberculosis [62], hepatitis B [63] and HIV [64]. However, all pELISA systems are designed for one analyte, and despite its great potential there are no multiplexed pELISAs thus far.

### 3.2. Multiplexed Colorimetric Detection

The nanoparticle-based colorimetric assays are one of the most attractive methods for the detection of various biomolecules such as DNA, RNA, enzymes, proteins, and other small molecules. The key parameter for colorimetric sensing is the ability to change colour of colloidal solution due to changes of noble metal nanoparticles size or distance between them. Such colorimetric sensors can be divided into four types: aggregation, etching, growth and nanoenzyme [65]. The most commonly used gold or silver nanoparticle-based sensors are of the aggregation type, in which optical features of solutions change depending on their level of aggregation. During aggregation of nanoparticles, the surface plasmons of the particles couple and their LSPR profile changes. The main advantages of colorimetric assays are that results can be monitored with the naked eye without the need for any instrumentation. Additionally, this method is simple, convenient, and low cost. Most studies have developed colorimetric detection tools for one analyte [66], however, some multiplexed colorimetric assays have been published.

Mancuso et al. used AuNPs and AgNPs for Kaposi’s sarcoma associated herpesvirus and *Bartonella* DNA simultaneous detection [67]. Specific DNA primers, which recognise targeted oligonucleotide sequences, were attached to AuNPs and AgNPs. When targeted oligonucleotides appear in solution, nanoparticles conjugated with primers bond with them and with each other. Consequently, aggregates of AuNPs and AgNPs are formed and the solution changes colour. Changes were identified visually and by measuring the absorption of the solutions. Authors have demonstrated that the proposed method works when AuNPs and AgNPs are mixed in the same solution—both colour change reactions can be seen independently of each other. In this assay, the limit of DNA detection for the AuNPs is 2 nM and for the AgNPs is 1 nM [67].

Heo et al. demonstrated two-plexed colorimetric assay using three types of nanoparticles, functionalised with DNA: gold nanoparticles (AuNP-DNA), silver nanoparticles (AgNP-DNA), and gold nanorods (AuNR-DNA) [68]. The mixture of these nanoparticles is black-coloured, but after targeted analytes are introduced, the colour of the solution changes, based on which type of nanoparticles (or combination) have aggregated. The authors used a modified CMYK (cyan, magenta, yellow, and key (black)) colour model: when one type of nanoparticle aggregates, corresponding colours are detracted from the mixture. For example, when red-coloured AuNPs aggregate due to interaction with targeted analytes, the mixture turns light-green (cyan and yellow combination). Using this approach, the authors successfully demonstrated the detection of thrombin and platelet-derived growth factor (PDGF) in human blood plasma. The sensitivity of this method was not high (the limit of detection of PDGF was 19 nM), however, it was possible to visually detect the detection limit and concentration (20 nM) with the naked eye [68].

Different approaches for the multiplexed colorimetric diagnostic detection of cancer was demonstrated by Di et al. [69]. The authors used decorated AuNPs and antibody conjugated exosomes for a nanozyme-assisted immunosorbent assay to detect CD63, CEA, GPC-3, PD-L1 and HER2 exosomal proteins from four cell lines as well as from clinical serum samples. This method allows users to differentiate the levels of the various proteins without additional labelling with detection antibodies, enabling the development of a quicker and much simpler testing procedure [69].

### 3.3. Multiplex SERS Imaging

Surface-enhanced Raman scattering (SERS) is a method for precise molecular detection based on the enhanced Raman scattering of biomolecules, which are localised on nanostructured SERS-active gold or silver surfaces. The advantages of this method are very high sensitivity down to the single molecule level, and the sharp molecularly specific spectra. The use of SERS instead of QD labelling for imaging is superior because SERS provides greater sensitivity for molecular analysis [70]. Moreover, the Raman signals exhibit much narrower peaks compared with traditional fluorophores and QDs; generally SERS signals are only 1–2 nm width [71]. Additionally, Raman signals are exceptionally stable and resistant to photobleaching, independently from measurement conditions. Thus, SERS tags are absolutely applicable for the multiplex detection of soluble biomarkers as well as multiplexed SERS-based cellular imaging (Figure 5).

Lee et al. (2012) demonstrated SERS-based cellular imaging technique to detect and quantify multiple breast cancer biomarkers expressed on plasma membranes [72]. Silica-encapsulated AuNPs were conjugated with CD24 and CD44 antibodies and fluorescent dyes (FITC and RuITC), creating dual mode nanoprobes (SERS and fluorescence). They demonstrated that simultaneous detection of CD24 and CD44 biomarkers in breast cancer cells (MDA-MB-231) can be enhanced by combining the two methods: fluorescence imaging for quick detection and SERS imaging for accurate localisation of biomarkers.

Later studies by the same group applied a SERS-based imaging method for the multicolour detection of three breast cancer cell biomarkers [73]. This study was caried out in human breast cancer cell lines (MDA-MB-468, KPL4 and SK-BR-3) by measuring the expression of EGF, ErbB2, andIGF-1 biomarkers. After visualisation, SERS-mapping images were recorded, which allowed them to localise and quantify EGFR, ErbB2 and IGF-1R proteins in cells. Analysis of the experimental results enabled identification and phenotyping of cancer cells [73].

Sun et al. (2015) used polydopamine encapsulated SERS probes for the detection of tumour-associated biomarkers (VEGF, EGFR and Vimentin) in different prostate cancer cell lines (LnCAP, PC-3, DU145) [74]. By using SERS images, the expression of three tumour biomarkers on the surface of prostate cancer cells were evaluated visually and three prostate cancer cell lines had been distinguished from each other. The authors suggest that usage of the multiplexed SERS imaging technique could improve the identification of cancer cell phenotypes [74].

Li et al. (2018) developed a multiplexed nanobiosensor by adapting the sandwich-type immunoassay and SERS methods for the detection of cancer biomarkers. Three soluble cancer protein biomarkers (soluble programmed death 1 (sPD-1), soluble programmed death-ligand 1 (sPD-L1) and soluble epithermal growth factor receptor (sEGFR)) were analysed directly from human serum [75]. In this study, for Raman signal improvement, gold-silver alloy nanoboxes were used as SERS tags to facilitate highly sensitive detection. The limit of detection for sPD-1, sPD-L1, and sEGFR achieved by this platform was 6.17 pg/mL, 0.68 pg/mL, and 69.86 pg/mL, respectively. They proposed that their nanobiosensor has huge potential in the clinic, because it can accurately and specifically detect several cancer biomarkers in human serum at once [75].

Another study demonstrated multiplex profiling of oestrogen receptor (ER), progesterone receptor (PR), and epidermal growth factor receptor (EGFR) expression in breast cancer tissue and normal tissue sections, using AuNP-based SERS tags [76]. 60 nm AuNPs with incorporated alkyne and nitrile groups and conjugated to primary antibodies against the growth factors were used as SERS tags. They demonstrated that their SERS nanotags have individual and definite bands in cellular regions without any Raman signal. The method was tested in vitro using the MCF-7 cell line, which expresses ER, EGFR, and PR at high levels and the normal cell line 3T3 in which expression of the biomarkers is downregulated. The results corresponded with data obtained with other methods. Additionally, the authors examined the use of multiple SERS tags for ER, EGFR, and PR imaging in human breast cancer tissue specimens. The results showed higher expression levels of these three biomarkers in breast cancer tissues compared to healthy samples, consistent with the diagnostic and prognostic classification of these tissues. Thus, the authors suggest that their proposed method has potential for multiplex imaging in clinical cancer diagnosis [76].

SERS is a reliable analytical method suitable for the detection of multiple analytes in biological samples even if there are extremely low quantities of target biomarker. By using SERS-nanotags, assays can be conducted directly in biopsy samples as well as in tissues or live cells, if necessary. However, utilisation of noble metal nanoparticles in multiplexed SERS biosensors is not yet cost effective and has limitations, such as high-cost fabrication of nanoparticles, low batch-to-batch consistency, high complexity, and relatively low specificity. Reproducible methodologies suitable for the mass production of testing kits based on gold or silver SERS nanotags also remain a challenge. Additionally, the majority of published multiplexed SERS imaging techniques require specific equipment, thus it would be complicated to adapt these methods for clinical needs. Despite these challenges, the field of multiplex SERS has great potential for innovation in diagnostic applications.

### 3.4. Plasmon-Enhanced Multiplexed Biosensing

Plasmon-enhanced fluorescence (PEF) is a phenomenon by which the fluorescence intensity of a nearby fluorophore can be remarkably enhanced by a plasmonic nanostructure [77]. By utilising PEF, the quantum efficiency and photostability of fluorophores are increased. Additionally, extremely low amounts of fluorophore could be detected. Thus, these features enable sensitive detection for very low abundance biomarkers. For the past few years, PEF-based biosensors have received a great deal of attention for the ultrasensitive detection of single analytes. Ventura et al. used the surface of patterned AuNPs for FITC fluorescence enhancement to detect immunoglobulins in real urine samples and showed that their method works in a 10–100 µg/L detection range with a limit of detection of 8 µg/L [78]. Zhang et al. used an Au nanohole array for prostate-specific antigen detection and demonstrated a limit of detection of 140 fM [79].

Wang et al. demonstrated thrombin and platelet-derived growth factor-BB (PDGF-BB) simultaneous detection using aptamer-modified AgNPs as a capture substrate, and fluorescent dye-modified aptamers as detection probes in a sandwich immunoassay [80]. The linear range of thrombin detection was from 55.6 pM to 13.5 nM, with the limit of 6.2 pM. PDGF-BB was detected with the range of concentration from 625 pM to 20 nM and the detection limit was 156 pM. They showed that by using AgNPs probes for Cy3 and Cy5 fluorescence enhancement, the detection limit could be improved 80-fold for thrombin and 8-fold for PDGF-BB, when compared to aptamers without enhancement [80].

Liu et al. developed a multiplexed antibody microarray for circulating biomarkers associated with lung cancer detection [81]. Microarray plates were modified with gold nanostructures to enhance the fluorescence of IRDye800, which was used to label detection antibodies in sandwich immunoassay. Lung cancer biomarkers CEA, CYFRA 21-1 and NSE were detected directly from human serum. Simultaneous detection of three biomarkers was achieved by printing capture antibodies onto 3 × 3 spot matrices. This approach enables a high-throughput testing of lung cancer biomarkers and improved specificity and sensitivity compared with convenient methods such as ELISA and Luminex assay [81]. Similar studies published by the same group demonstrated detection of diabetes biomarkers [82] as well as zika and dengue viruses [83].

Min et al. demonstrated a new and simple technique that allows the multiplexed detection of biomarkers in extracellular vesicles [84]. In this study, plasmon-enhancement was achieved by using Au nanoholes as a substrate. Firstly, vesicles were captured on Au nanoholes by a biotin-avidin reaction. Then, vesicles were stained with antibodies conjugated with fluorescent labels (Alexa Fluor 488, Cy3, Cy5, Cy5.5). Au nanoholes operated as amplifiers for fluorophores under excitation. They applied the designed assay to detect glioblastoma biomarkers CD-pan (CD9, CD63, and CD81), with EGFR, EGFRvIII and GAPDH as control markers, from supernatants of glioblastoma cells. Fluorescence signals of multiple fluorophores were amplified by one order, and both transmembrane and intravesicular biomarkers were detected at the single extracellular vesicle level. However, the proposed assay still has limited multiplexing capability, which is dependent on fluorescent microscope systems, allowing the visualisation of up to three or four fluorophores in one sample. Additionally, this system was not tested with clinical samples, thus its real applicability still is unknown [84].

Liu et al. developed an assay for the detection of SARS-CoV-2 antibodies by using nanostructured plasmonic gold substrate as a near-infrared fluorescence amplifier [85]. In this study, IgG and IgM antibodies against SARS-CoV-2 were detected directly from human serum and saliva using a sandwich immunoassay in microarray plate. They demonstrated high specificity and sensitivity for the detection of antibodies against SARS-CoV-2. Additionally, antibodies to viruses associated with common colds did not cross-react with SARS-CoV-2 or SARS, or reaction was low. Application of this fast and sensitive testing method in daily clinical practice would allow population-based diagnostic mass screening of COVID-19 [85].

## 4. Upconverting Nanoparticles

Low background signal and high specific signals from molecules under investigation are two main criteria for precise and effective spectral encoding for multiplexed analysis of various biosamples [86]. The main drawback of previously described methods is interference between coded signals and reporter signals at low concentrations, which leads to inaccurate test results. Nanoparticles are mostly optically active in UV–VIS region, where many biomolecules also show strong autofluorescence (Figure 6a). In this case, upconverting nanoparticles (UCNPs) could be useful for in vivo biosensing. UCNPs are inorganic nanocrystals, commonly rare-earth-doped nanoparticles (RENPs), that allow for the generation of anti-Stokes emission under near-infrared (NIR) excitation. Upconversion (UC) emission itself is a unique process, where low energy NIR light is converted into higher energy light through the sequential absorption of multiple photons or energy transfer. UCNPs exhibit great properties with sharp emission bands, large anti-Stokes shift, high resistance to photobleaching, long lifetimes (~ms), high detection sensitivity, low toxicity, and no interference from biomolecular autofluorescence [87] (Figure 6b). The popularity of UCNP application in various biomedical research is constantly growing, including the development of nanobiosensors.

Gorris et al. described UCNPs as background-free codes for multiplexed analyses [86]. In the system proposed by the authors, the surface of the UCNPs (NaYF_4_:Yb,Er and NaYF_4_:Yb,Tm) is modified with a screen layer which consists of different amounts of an organic dye. The absorption spectrum of this dye overlaps with one of the UCNP bands, thus the emission from this band is re-absorbed to different levels by the organic dye. Another emission band is used as a reference. These UCNPs and organic dye complexes could be used as coding elements which provide ratiometric codes in multiplexed system [88].

Zhang et al. created upconversion nanobarcodes (UPNBs)—uniform Y_2_O_3_:Yb,Ho,Tm@Y_2_O_3_@SiO_2_(COO-) core/shell nanoparticles, which could be used for optical encoding in multiplexed immunoassays [89]. The authors demonstrated that by changing the composition of nanoparticles, several different UPNBs can be created because each nanobarcode has an individual spectrum. Additionally, in order to test synthesised nanobarcodes, UPNBs were conjugated with mouse and rabbit antibodies for two-plexed immunoassays. Additionally, microspheres modified with goat anti-mouse and goat anti-rabbit antibodies (both cascade blue-labelled) were used for the immunodetection of potential receptors. In this way, the recognition process was simulated in the model system. Specific binding between primary antibodies conjugated to UPNBs and secondary antibodies conjugated to microspheres were observed, thus providing proof-of-principle of this model immunoassay [89].

Zhang et al. designed upconverting nanocrystals encoded magnetic microspheres (UCNMMs), which have the potential application for fast separation and multiplexed immunoassays [90]. Six unique upconverting nanocrystals with different upconversion emission spectra were synthesised by a solvent–thermal process. Both upconverting nanocrystals and magnetic nanoparticles were encapsulated into porous poly(styrene-co-EGDMA-co-MAA) beads to obtain UCNMMs. Later UCNMMs were conjugated with goat anti-mouse IgG. For the purpose of detection, FITC-labelled rabbit anti-mouse and IgG PE-labelled goat anti-rabbit IgG antibodies were used. The immunoassay results demonstrated no interrelation between upconversion emission, which was used for encoding, and fluorescence of dyes (FITC, PE), which was used as a reporter. Thus, the authors demonstrated the advantage of upconverting nanomaterials’ utilisation as barcodes in multiplexed immunoassays compared with traditional down-conversion materials (fluorescent dyes or QDs). The detection limit of mouse IgG in this model immunoassay was 0.01 ng/mL. Thus, their proposed detection system has huge potential applications in multiplexed immunoassays [90].

In order to improve the diagnostic detection of several viruses such as influenza (A and B), respiratory syncytial virus, and adenovirus, Kazakova et al. developed a multiplex serological microarray immunoassay for the simultaneous detection of serum IgG antibodies against these viruses [91]. Microarray plates were coated with streptavidin and the serum antigens as well as negative and positive controls for the samples. Different locations of each of the spotted antigens allowed them to implement specificity of this immunoassay. For detection purposes, anti-human IgG-coated NaYF_4_:Yb,Er upconverting nanoparticles were used. UCNPs were chosen to avoid autofluorescence signals from the sample and to increase detection sensitivity. The nanobiosensor was effectively used for the simultaneous detection of antibodies against seven different viruses. The authors suggest that their multiplexed immunoassay is a promising tool for the diagnostic detection of viral infections in serum samples, and can be utilised in epidemiological and seroprevalence studies [91].

Li et al. demonstrated multiplexed upconversion imaging in vivo by registering distinct lifetimes of UCNPs after excitation at 808 nm [92]. The authors used an NaYF_4_@NaYbF_4_@NaYF_4_:Yb/Tm@NaYF_4_ UCNP to control the NIR upconversion luminescence (UCL) lifetime in the range 78–2157 μs detected at 808 nm. They conducted in vivo multiplexing experiments using these UCNPs with tuneable lifetimes for in vivo imaging. For the temporal multiplexed imaging, UCNPs with three different lifetimes were injected into a Kunming mouse through tail vein injection and implemented subcutaneous injection into the left and right of the abdomen. UCL signals with two different lifetimes were observed in liver and abdomen subcutis, which corresponded well with UCL imaging. The authors suggest that their temporal upconversion application has the potential for use as a new optical multiplexed imaging technique [92].

Several studies have demonstrated application of RENPs for multiplexed in vivo imaging by utilising only down-conversion emission and lifetime imaging [93,94,95]. Fan et al. (2018) designed RENPs barcodes with different lifetime characteristics for multiplexed breast cancer biomarkers (ER, PR, HER2) detection in tumour bearing mice [93]. The expression of tumour biomarkers was quantified by lifetime imaging, and a recognition algorithm was used to resolve subtype of tumour, depending on the different expression of biomarkers.

## 5. Conclusions

In this review article, we have provided a comprehensive overview of the current state-of-the-art development of multiplexed nanobiosensors. We have paid special attention to how different nanomaterials could help to achieve better sensitivity and specificity in classical bioassays. The advantages are anticipated in the development of ELISA, flow cytometric, electrochemical and immunofluorescence techniques, with the combination of nanomaterials such as quantum dots, magnetic beads, gold, silver and upconverting nanoparticles. These nanomaterials applied in rapid and cost-effective multiplexed nanobiosensors show promise for the specific and highly sensitive detection of clinically relevant biomarkers. However, there are still some challenges that remain before nanobiosensors can be applied for daily diagnostic purposes. Firstly, colloidal stability of nanoparticles is usually not sufficient for longer shelf life. This disadvantage could be eliminated when nanoparticles are dispersed on substrates by chemical entrapment or by growing them directly on substrates. Another big issue is the biofunctionalisation of nanomaterials. Despite the availability of a broad variety of chemical methods of biomolecule attachment to the surface of nanoparticles, this process usually requires a controlled environment for long-term stability. Repeatability and sensitivity are also key factors for translation from laboratory to clinic. Laboratories are equipped with exclusive and specific equipment, which allows investigations to reach greater sensitivity in the developed assays. However, in clinical practice, such equipment would be expensive and time consuming. Nevertheless, several research projects are concentrating on developing simple, cheap and quick assays, suitable for clinical translation and wider implementation in the operational environment of the clinic. Various advantageous aspects of nanomaterials used in multiplexed nanobiosensors are summarised in Table 1, and current evidence implies that these nanomaterials could be used for the development of highly sensitive multiplexed biosensors.

## Figures and Tables

**Figure 1 sensors-20-06890-f001:**
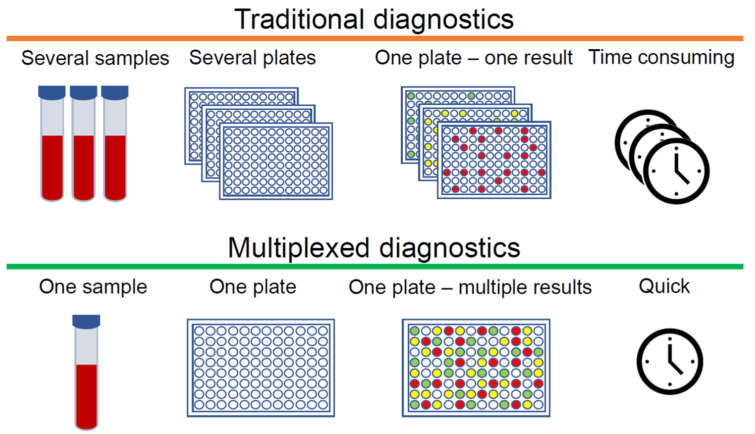
Differences between traditional and multiplexed diagnostics.

**Figure 2 sensors-20-06890-f002:**
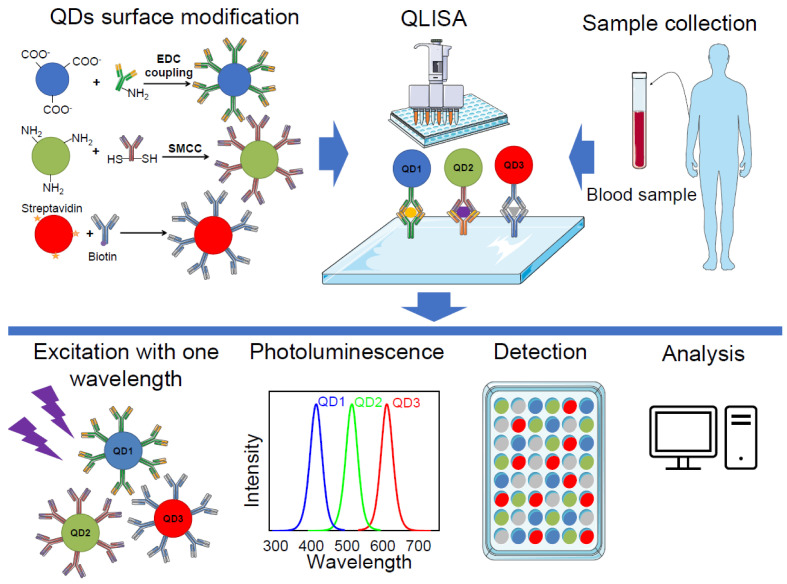
Multiplexed diagnostics using quantum dot-linked immunosorbent assay (QLISA). Firstly, the surface of the quantum dots (QDs) is modified with specific antibodies using 1-Ethyl-3-(3-dimethylaminopropyl)carbodiimide (EDC) coupling, succinimidyl-4-(N-maleimidomethyl) cyclohexane-1-carboxylate (SMCC) conjugation, or streptavidin–biotin interaction. Clinical samples from patients are processed and loaded to a QLISA plate, which is coated with capture antibodies. Then, modified QDs are loaded and the plate is scanned using a microplate reader. Different QDs in the sample are excited with the same wavelength and several bands of photoluminescence are detected. The final step is the analysis of the acquired data.

**Figure 3 sensors-20-06890-f003:**
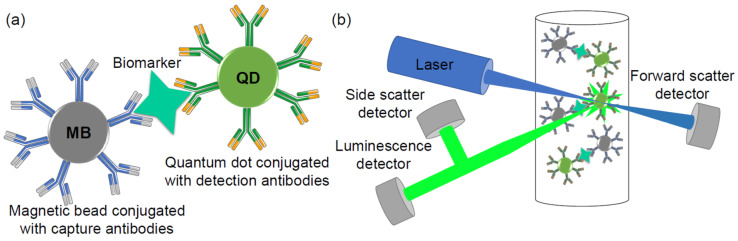
Schematic illustration of a magnetic bead–quantum dot (MB–QD) sandwich assay for biomarker capture and detection; (**a**) magnetic bead–biomarker–quantum dot conjugate; (**b**) simplified flow cytometer scheme.

**Figure 4 sensors-20-06890-f004:**
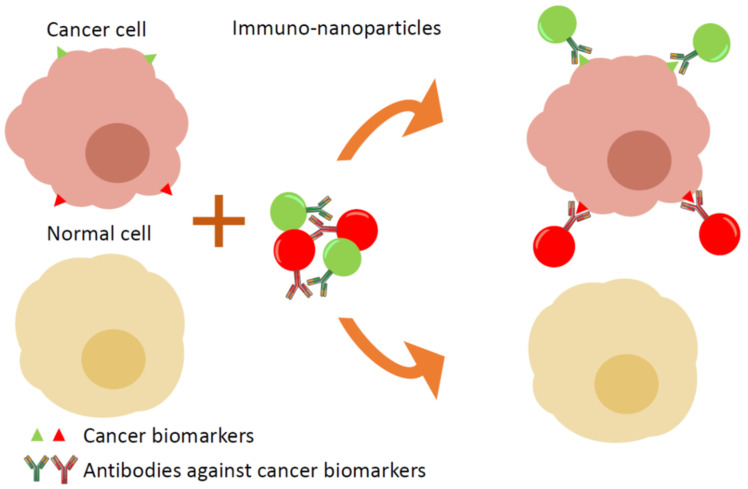
Targeting of cancer cells in tissue samples: multiple immuno-nanoparticles with specific antigens against cancer biomarkers label cancer cells. Each antibody is conjugated with different coloured luminescent nanoparticles, which allows the simultaneous detection of several biomarkers in one sample.

**Figure 5 sensors-20-06890-f005:**
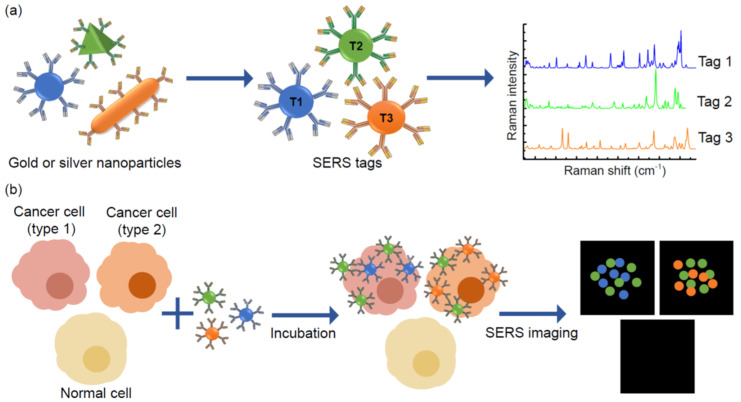
Principles of multiplexed detection using the surface-enhanced Raman scattering (SERS) technique. (**a**) Noble metal nanoparticles, that are different in size and shape, have unique Raman signals with narrow peaks, thus they could be used as SERS tags. (**b**) Schematic illustration of simultaneous detection of three tumour associated antigens expressed different types of cancer cells by SERS imaging.

**Figure 6 sensors-20-06890-f006:**
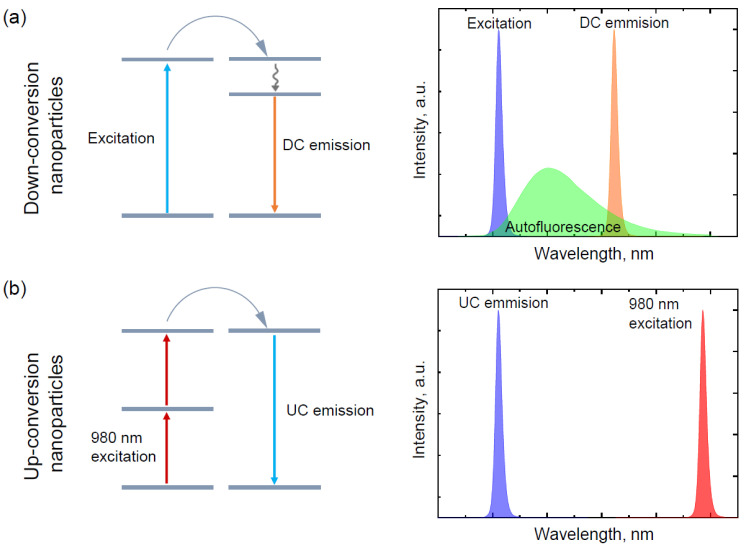
Schematic illustrations of underling luminescence mechanisms: (**a**) down-conversion nanoparticles are excited in the region where various biomolecules also could be excited and the emission of nanoparticles could overlap with autofluorescence; (**b**) up-conversion nanoparticles are excited in NIR region where biomolecules are not excited, thus there is no autofluorescence of the sample.

**Table 1 sensors-20-06890-t001:** Summary of described multiplexing nanobiosensors: used nanoparticles, method of detection, number of detected analytes, detection range and detection limit (if provided).

Nanoparticles	Method	Number of Analytes	Detection Range	Detection Limit	Reference
Quantum dots	QLISA	3	30–1000 ng/mL	30 ng/mL	[18]
Quantum dots	QLISA	3	0.05–10 ng/mL	5 pg/mL	[19]
Quantum dots	Magnetic bead–quantum dot assay	3	1.03–111 ng/mL 9.26–1000 ng/mL	0.19ng/mL 0.97ng/mL 0.37ng/mL	[30]
Quantum dots	Magnetic bead–quantum dot assay	3	3.9–125.0 ng/mL	364 pg/mL 38 pg/mL 370 pg/mL	[31]
Quantum dots	Magnetic bead–quantum dot assay	3	-	-	[32]
Quantum dots	Magnetic bead–quantum dot assay	5	-	0.01–0.06 ng/mL 0.06–1.5 KU/L	[33]
Quantum dots	Magnetic bead–quantum dot assay	5	-	0.01–0.02 IU/mL	[34]
Quantum dots	Flow cytometry	2	0.52–30 ng/mL 0.13–10 ng/mL	-	[35]
Quantum dots	Flow cytometry	3		7–10 ng/mL	[36]
Quantum dots	Electrochemical immunoassay	2	0.001–0.1 pg/mL	0.4 fg/mL	[40]
Quantum dots	Electrochemical immunoassay	2	10^−6^–1 U/mL 10^−4^–100 U/mL	0.1 μU/mL 10 μU/mL	[41]
Quantum dots	Immunocytochemistry	2	-	-	[42]
Quantum dots	Immunocytochemistry Immunohistochemistry	5	-	-	[43]
Quantum dots	Immunohistochemistry	4	-	-	[44]
Quantum dots	Immunohistochemistry	2	-	-	[45]
Quantum dots	Immunochromatographic dipstick method	2	1–50 ng/mL 5–50 ng/mL	37.8 pg/mL 42.6 pg/mL	[48]
Quantum dots	Lateral flow assay	2	1.3–480 ng/mL 2.8–680 ng/mL	0.16 ng/mL 0.35 ng/mL	[49]
Quantum dots	Microfluidic system	2	1.0–40 ng/mL	0.3 ng/mL 0.4 ng/mL	[50]
Gold nanorods	LSPR spectroscopy	2	-	-	[57]
Gold and silver nanoparticles	Colorimetric Spectroscopy	2	-	2 nM 1 nM	[67]
Gold and silver nanoparticles	Colorimetric Spectroscopy	2	15–40 nM	19 nM	[68]
Gold nanoparticle decorated exosomes	Colorimetric Spectroscopy	5	-	-	[69]
Silica encapsulated gold nanoparticles	SERS-based cell imaging	2	-	-	[72]
Silica encapsulated gold nanospheres	SERS-based cell imaging	3	-	-	[73]
Polydopamine encapsulated gold nanoparticles	SERS-based imaging	3	-	-	[74]
Gold-silver alloy nanoboxes	SERS	3	-	6.17 pg/mL 0.68 pg/mL 69.86 pg/mL	[75]
Gold nanoparticles	SERS-based imaging	3	-	-	[76]
Gold nanoparticles	SERS	2	0.5–100 ng/mL	0.41 ng/mL 0.35 ng/mL	[96]
Gold nanoparticles	SERS	2	-	-	[97]
Silver nanoparticles	PEF-based immunoassay	2	0.0556–13.5 nM 0.625–20 nM	6.2 pM 156 pM	[80]
Gold nanoislands	PEF-based immunoassay	3	0.1–100 ng/mL	0.05 ng/mL 0.15 ng/mL 0.19 ng/mL	[96]
Gold nanoholes	PEF-based fluorescence microscopy	4	-	-	[84]
Upconverting nanoparticles	Spectroscopy	2	-	-	[88]
Upconverting nanoparticles	Confocal luminescence imaging	6	-	-	[89]
Upconverting nanocrystals encoded magnetic microspheres	Fluorescence microscopy Flow cytometry	Shown 2 Potentially 6	-	0.01 ng/mL	[90]
Upconverting nanoparticles	Microarray immunoassay	7	-	-	[91]
Upconverting nanoparticles	UCL intensity and lifetime imaging	3	-	-	[92]
RENPs	Lifetime imaging	3	-	-	[93]
Gold nanoparticles	Flow cytometry	3	0.2–20 nM	0.4 nM 0.1 nM 0.2 nM	[98]
Silver nanoclusters	Spectroscopy	2	10–100 nM	2.4 nM 5.6 nM	[99]

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
