# Peer review of "Multiplexed Nanobiosensors: Current Trends in Early Diagnostics"

_sensors, 2020, doi:10.3390/s20236890_

Round 1

Reviewer 1 Report

Jarockyte et al. present a minireview on the utility of certain types of nanoparticles for multiplexed sensing. Indeed, the use of nanoparticles for sensing and diagnostic applications is a timely research topic with potential in the biomedical field. The authors cover several types of nanoparticles and sensing methods, with a particular emphasis on quantum dots, giving selected illustrative examples in each case. Therefore, the manuscript could be suitable for publication in Sensors if the authors address the following points:

1.- The term nanobiosensors is ambiguous. The authors are actually reporting the use of nanoparticles for multiplexed sensing coupled with different readout methods – however, sensing still requires the use of large (not nanoscale) equipment. Although using the term nanobiosensors could still be accepted, the authors should introduce their definition of nanobiosensors and clarify this point to the reader.

2.- Using the term noble metal nanoparticles (in the abstract, headings and manuscript) is rather broad and confusing as the authors are actually referring gold and silver nanoparticles. I feel the authors should refer to them just as gold and silver nanoparticles or to include other examples of particles in this category.

3.- Line 214: there is a typo (two consecutive dots)

4.- Gold and silver nanoparticles have been used for multiplexed colorimetric sensing, based for instance in analyte-induced aggregation and subsequent color change, with a number of studies that can be found in the literature. I recommend the authors should also refer to this and include some examples.

4.- A reflection on the current challenges to address and expected evolution of the field would be appreciated (for instance, in the conclusions section).

Reviewer 2 Report

The topic addressed in this review is timely since the multiplexed detection of analytes is a major goal in biosensing and one way to achieve it is by using the unique features of the nanoparticles. The authors focus their manuscript on the applications of multiplexing to medical diagnostics, that is a field in which the detection of a panel of analytes is nowadays recognized as the most appropriate strategy to carry out a precise diagnosis. The nanoparticles are considered in three possible versions: quantum dots, metal nanoparticles and nanoparticles for upconversion. In all the cases, the crucial steps towards an application are outlined and a synoptic table is proposed to summarize the main results.

In my view, the manuscript deserves publication, but after the following comments are addressed.

Major comments.

1) The section 3 lacks a subsection concerning the Plasmon (or Metal) Enhanced Fluorescence that can be considered even more practical than SERS. This is a topic that should not be overlooked in a review like this. Apart from taking inspiration from reviews like those by M. Bauch et al (Plasmonics 9, 781-799 (2014)) and Y. Jeong et al (Biosensors and Bioelectronics 111, 102-116 (2018)), the authors might consider some recent advances like those reported by S. Pawar et al (ACS Omega 4, 5983–5990 (2019)) and B. Della Ventura et al (ACS Appl. Mater. Interfaces 11, 3753–3762 (2019)).

2) Line 60-62. The sentence reported in this line seems to refer to actual devices or kits commercially available. If the sentence was really true even the research would not be so active in the field. The authors are invited to clarify what they mean.

3) The scheme reported in Figure 3 does not suggest any multiplexing measurement and should be modified accordingly.

Minor comments.

The text should be revised. Some mistakes include but are not limited to the following examples.

1) Line 159. The limit of detection is 5 pg/mL.

2) Line 176. The sentence is not clear. Is “whereas” missing after the second coma?

Reviewer 3 Report

The review paper « Multiplexed nanobiosensors: Current trends in early diagnostics” by Jarockyte et al. describes diagnostic tools for detection of biomarkers of clinical significance in a multiplex way. A special attention was taken to biosensors based on quantum dots, noble and rare earth metal nanoparticles.

The manuscript is very clear and easy to follow. However, some major modifications are needed:

- the authors should give a more detailed description and explanation of noble metal plasmonics.

- for QD immunosensors, the authors should make a comparison between conventional sensors and QD biosensors, such as the detection methods and detectable signals.

-paper based multiplex nanobiosensors should be presented too. Such section might be helpful for many researchers working on paper based biosensors.

-It is true that nanobiosensors like SERS, SPR etc have a relatively low instrument cost, but the utilization of nanoparticles that are frequently characterized by the low durability raise the cost of using plasmonics biosensors. The authors are suggested to add some discussions on this.

"Conclusion should contain a discussion about the limitations of presented biosensors. No method is perfect, and the grey sides of multiplex nanobiosensors should be presented. Moreover, the limitations should be discussed at the end of different sub-sections, and an overall picture should be presented in the final section as well.

Round 2

Reviewer 2 Report

The authors revised the manuscript and addressed all the comments raised by the reviewers.

Reviewer 3 Report

A very nice review. Accept in the present form.